# Hostility has a trivial effect on persuasiveness of rebutting science denialism on social media

Philipp Schmid [1,2,3✉] & Benedikt Werner[4]

Polarised social media debates between science deniers and advocates for science frequently devolve into hostilities. We conducted four preregistered experiments ($N = 3226$; U.S. Americans) to assess how hostility influences the impact of misinformation from science deniers and rebuttals from advocates in social media discussions about vaccination (Experiment 1–3) and genetically modified food (Experiment 4). Results revealed only trivial to small effects of hostility on the persuasiveness of discussants: most internal meta-analyses revealed that the effects of hostility were smaller than the smallest effect size of interest (SESOI: $d = 0.2$). Thus, rebuttal is effective in mitigating the impact of misinformation on attitudes towards behaviours dismissed by deniers (for example, vaccination) and intentions to perform these behaviours, even if advocates for science lose their temper. Not responding had negative effects. Likewise, misinformation was impactful even if presented in a hostile tone. Hostility, however, consistently reduced the perceived competence of messages.

[1] Institute for Planetary Health Behaviour, University of Erfurt, Erfurt, Germany. [2] Centre for Language Studies, Radboud University, Nijmegen, The Netherlands. [3] Health Communication, Department of Implementation Research, Bernhard-Nocht-Institute for Tropical Medicine, Hamburg, Germany. [4] Department for Applied Microeconomics, University of Erfurt, Erfurt, Germany. ✉email: philipp.schmid@uni-erfurt.de

The fact of man-made climate change, the efficacy of recommended vaccines, or the safety of approved genetically modified (GM) foods are just a few examples of robust scientific findings that have been questioned by misinformation around the world. These types of misinformation—expressions of dismissal of well-established scientific evidence—are expressions of science denialism[1–3]. Science deniers reject scientific evidence because it poses a threat to their economic, social, or psychological interests[3,4]. For example, spreading doubts about climate change can ensure fossil fuel sales, sharing the beliefs of a flat earth society can satisfy a need for belongingness, and refusing all vaccines or GM organisms can be a strategy to cope with fears of needles or new technologies. In the face of scientific evidence, science deniers employ a variety of different strategies to justify their denial. They cite fake experts, conduct logical fallacies, raise impossible expectations, cherry pick data and construe conspiracy theories[5,6]. These strategies are also often used in misinformation to persuade others[7]. Exposure to misinformation that denies science can reduce the public's positive attitudes toward behaviours dismissed by science deniers (e.g., vaccination) and the intention to perform these behaviours[7,8]. Advocates for science can mitigate the damage of science denialism by rebutting misinformation in public discussions, for example, on social media[9–12]. Advocates for science "follow scientific consensus and argue for the evidence-based position"[9]. They engage in science communication and rebuttal of misinformation either as professionals (e.g., scientists, science journalists) or as knowledgeable laypeople (e.g., amateur fact-checkers).

Due to the highly polarised nature of some of the scientific issues, misinformation and rebuttals are often accompanied by offensive and aggressive language[13–15]. For example, during the COVID-19 pandemic, the dehumanising term "sheep" was used in misinformation to describe people who follow vaccination recommendations, while the term "covidiots" was used to attack the integrity of unvaccinated individuals[16,17]. The use of such intimidating terms in public discussions can be defined as a form of hostility[18] aimed at expressing anger but also intentionally silencing other discussants[19,20]. Hostility, however, may have unintended consequences for the persuasiveness of the hostile discussant. That is, hostility may be a double-edged sword in social media discussions about scientific issues. In this study, we analyse whether the persuasive impact of misinformation that denies science and rebuttal decreases when the social media discussion heats up.

Hostile language tends to be perceived negatively by audiences in various public contexts such as work settings[21], in political campaigning[22] and in communication about scientific issues[23,24]. Referring to the latter, negative perceptions may be due to the fact that hostility is perceived as incompatible with the image of a scientifically competent discussant[25–28]. In line with established persuasion theories, the perceived incompetence of a discussant can serve as a simple rejection cue or even as a relevant argument for not trusting the discussant on the matter at hand (elaboration likelihood model[29]; heuristic-systematic model[30]). Thus, we predicted that by using hostile language, science deniers and advocates for science deteriorate their standing as a competent discussant and thereby damage their own persuasive power in social media discussions.

Expectancy violations theory suggests that rebuttals may lose persuasive power by using hostile language for another reason, namely, by violating the audience's expectancies in public discussions[31]. According to the expectancy violations theory, persuasive attempts are least effective if they violate the audience's expectancies in a negative way and most effective if they violate the audience's expectancies in a positive way. Expectancies can be based on social norms but can also be influenced by situational cues[32]. In public discussions, the hostility of the previous speaker can act as such a situational cue[33]. Thus, a hostile rebuttal following a neutral misinformation may be a negative violation of expectancies because the audience might not expect hostility from an advocate for science without previous signs of provocation. In contrast, a hostile rebuttal following an equally hostile misinformation may come across as appropriate or understandable. In the same vein, a neutral rebuttal after a hostile misinformation may even be perceived as a positive violation of expectancies because people might not expect decency in the face of hostility. Thus, we expected that expectancy violations are another mechanism by which hostility reduces the persuasive power of rebuttals in social media discussions.

Despite the expected inferiority of hostile rebuttals over neutral rebuttals, hostile rebuttals may nevertheless contain strong counterarguments that can mitigate the damage of a denier's misinformation. That is, a hostile rebuttal may still be better than leaving the misinformation entirely unchallenged. In fact, research shows that it is more important to rebut misinformation than to worry about the right format[9,34]. Thus, we predict that hostility may weaken but not neutralise the benefit of rebuttal in social media discussions.

Furthermore, there is reason to believe that the described potential effects of hostility are also a function of the characteristics of the audience. First, messages that are insensitive to others' feelings have been shown to make communicators appear more authentic, especially among audiences that sympathise with the speaker[35]. In turn, an audience's sympathy towards a speaker is driven by similarity to them[36,37]. Thus, the use of aggressive language may not reduce the persuasive power of a message when used among audiences who themselves are verbally aggressive. Second, following dual-process theories of persuasion (e.g., elaboration likelihood model[38]), hostility can be interpreted as a simple rejection cue but it may also be interpreted as entirely irrelevant by an audience that primarily focuses on argument strength. Individuals usually focus more on argument strength than on peripheral cues if they are highly involved in the discussed issue and if they have a general high need for cognition[39,40]. Thus, the impact of hostility may be a function of the audience's need for cognition and issue involvement. Lastly, social media seems to be a convenient space for the spread of hostility[41] and individuals who are frequent social media users may react differently to hostile content than individuals who are relatively new to social media. Thus, we explored whether the impact of hostility is a function of the audience's frequency of social media use.

## Methods

**General information.** We conducted four preregistered experiments to analyse the impact of hostility in the specific context of social media discussions involving misinformation about vaccination and GM foods. Vaccination and GM organisms are highly promising technologies to fight major threats to global public health, such as infectious diseases and malnutrition. While scientists argue that risks and benefit analyses are crucial for each vaccine and GM food product, science deniers reject both technologies a priori and fuel general vaccine hesitancy and fears about GM organisms in the public[1,6,9]. Identifying effective interventions against misinformation that work across topics is useful to support the work of advocates for science. In addition, testing interventions across these topics provides an actual challenge for the generalisability of hypothesised effects because both topics have several unique aspects for the individual decision-maker.

Participants read a fictitious social media discussion between a science denier and an advocate for science in all experiments. In these discussions, the denier shares misinformation while the advocate counterargues against the misinformation by uncovering what is misleading about the misinformation (i.e., refutational messaging) and by providing additional scientific facts[9]. By engaging in this form of rebuttal, the advocate advocates for the scientific perspective (e.g., the safety of vaccination). We assessed individuals' attitudes towards a behaviour dismissed by science deniers and the intention to perform this behaviour before and after the social media discussion. These primary outcomes are central in research of persuasion psychology and have been used in previous studies about rebutting science denialism in public discussions[9,42]. In Experiments 1–3, we focused on vaccination as a topic of the social media discussion. In Experiment 1, we tested the impact of a hostile denier and a hostile advocate in a 2 (hostile denier versus neutral denier; between subjects) × 2 (hostile advocate versus neutral advocate; between subjects) × 2 (time of measurement: before versus after the debate; within subjects) mixed design. In Experiment 2, we replaced the neutral advocate condition with an advocate absent condition to test if hostile rebuttals are more effective than leaving the stage to the denier. In Experiment 3, we combined all conditions from Experiments 1 and 2 as a replication attempt of the initial findings. In Experiment 4, we used the full design from Experiment 3 but changed the topic of the social media discussion to increase the generalisability of findings. That is, in Experiment 4, we focused on the safety of genetically modified foods. An overview of designs and experiment characteristics is provided in Table 1. All experiments were preregistered. Preregistration protocols are available at aspredicted.org (Experiment 1, 09/22/2020: https://aspredicted.org/vp626.pdf; Experiment 2, 03/04/2021: https://aspredicted.org/7zy4r.pdf; Experiment 3, 06/18/2021: https://aspredicted.org/7ui2w.pdf; Experiment 4, 06/23/2021: https://aspredicted.org/tv3sw.pdf). The internal meta-analyses, the manipulation checks, and the equivalence tests were not preregistered. The experiments were conducted from 09/22/2020 until 07/16/2021 (Experiment 1: 09/22/2020–10/16/2020; Experiment 2; 03/08/2021–04/01/2021; Experiment 3: 06/18/2021–07/01/2021; Experiment 4: 06/23/2021–07/16/2021).

All experiments conform to the ethical principles for psychological research provided by the German Research Foundation. They are negligible-risk research and involve only nonidentifiable data about human beings and were therefore exempt from the requirement for ethical approval by the Institutional Review Board of the University of Erfurt. Participants gave their informed consent, could quit the experiments at any time, and received a debriefing at the end of the experiment. All individuals who have fulfilled the criteria for authorship required by Nature Portfolio journals are listed as authors of this article.

**Participants**. All experiments were conducted online using Prolific.co for recruiting. Convenience sampling was used for all experiments. Participation was a voluntary decision and participants could quit the survey at any time. Therefore, individuals intrinsically interested in the topic of the experiments could have been more willing to finalise the study. We tried to reduce this potential bias with adequate compensation of participants. Participants received a compensation of £1.63 (Experiment 1) or £1.25 (Experiments 2–4). The payment was in line with the recommended hourly rate of £7.50. $N = 592$ participants clicked on the link of Experiment 1, 541 proceeded after the introduction page and 521 finished the experiment. $N = 353$ participants clicked on the link of Experiment 2, 324 proceeded after the

introduction page and 310 finished the experiment. $N = 1351$ clicked on the link of Experiment 3, 1249 proceeded after the introduction page and 1200 finished the experiment. $N = 1427$ clicked on the link of Experiment 4, 1277 proceeded after the introduction page and 1195 finished the experiment. In total, $N = 3226$ U.S. adults completed the experiments (Experiment 1: $n = 521$, Experiment 2: $n = 310$, Experiment 3: $n = 1200$, Experiment 4: $n = 1195$). Information on the gender, age, and education of the participants is provided in Table 1. Information on gender, age, and education was provided by participants. Data on ethnicity was not collected. U.S. online users were considered a relevant target group for the research questions because the study focuses on the impact of hostile language in online environments and hostility is frequently experienced in online discussions in the US.

Sample sizes were determined via a priori simulation-based power analyses[43]. For Experiment 1, we aimed for a statistical power of at least 0.8 to detect the hypothesised main effects and pairwise comparisons in a 2×2-between-subjects-ANOVA, given $\alpha = 0.05$ and assumed effect sizes of $d \geq 0.35$ (informed by similar experiments involving civility[44,45]). For Experiment 2, the targeted smallest effect size for the main effects was $d = 0.37$ based on similar experiments involving rebuttal vs. no rebuttal comparisons[9]. In Experiments 3 and 4, we aimed to detect $d \geq 0.2$ for the main effects as the smallest effect size of interest. Participants were excluded from participation in subsequent experiments if they participated in a previous experiment. Thus, all measurements were taken from distinct samples. All data from participants were included in the primary analysis without any exclusions. Exploratory robustness analyses excluding specific participants (e.g., speeders) are provided below.

**Measures**. Primary dependent variables were attitudes towards a behaviour dismissed by science deniers and the intention to perform this behaviour. These primary outcomes were adapted from previous studies about rebutting science denialism in public discussions[9,42]. Participants also rated the denier's and the advocate's comments with regard to how polite, competent, context-typical, context-expected and positive/negative they had perceived them. The competence measure served as the mediator for the competence hypothesis, while the latter three measures were used to assess expectancy violations in Experiment 1. We removed the items for measuring expectancy violations in Experiments 2–4 because we dropped the expectancy violation hypothesis. The measure of politeness served as a manipulation check of the hostility manipulation (see manipulation and robustness checks below). Furthermore, potential moderator variables were measured: In Experiments 1 and 2, participants filled out a short verbal aggression subscale from the Brief Aggression Questionnaire[46] and indicated their frequency of social media use. In Experiments 3 and 4, we measured verbal aggression with a 10-item version of the verbal aggressiveness scale[47,48]. We also measured need for cognition[49], issue involvement[50], and social desirability[51]. Finally, attention check items were implemented in all experiments. Supplementary Tables 1–4 present means and standard deviations of all primary outcome measures and moderator variables of all experimental conditions. Supplementary Table 5 presents the full list of assessed variables, wordings of items, reliability analyses and references.

**Statistical analyses**. Data from all experiments in the study were collected using the web-based Enterprise Feedback Suite (EFS) by Tivian and stored and analysed on a computer. We used the statistical software R version 4.2.2 and the following additional

**Table 1 Overview of study characteristics of all experiments.**

| Experiment | Design | Primary outcomes | Mediators of experimental factors on outcomes | Moderators of effects of experimental factors on outcomes | Sample size and characteristics | Scenario |
|---|---|---|---|---|---|---|
| 1 | 2 (hostile denier vs. neutral denier; between subjects) × 2 (hostile advocate vs. neutral advocate; between subjects) × 2 (time of measurement: before vs. after the debate; within subjects) mixed design | Attitude towards vaccination against fictitious disease (dysomeria) Intention to get vaccinated against (dysomeria) | Perceived expectancy of denier's or advocate's message Perceived competence of denier's or advocate's message | Verbal aggressiveness Frequency of social media use | N = 521; U.S. adults; online sample (Prolific); $M_{Age}$ = 34.05, $SD_{Age}$ = 10.46; n = 248 women, n = 269 men, n = 3 non-binary participants; n = 14 high, n = 504 middle education | Social media discussion about vaccination against fictitious disease |
| 2 | 2 (hostile denier vs. neutral denier; between subjects) × 2 (hostile advocate vs. advocate absent; between subjects) × 2 (time of measurement: before vs. after the debate; within subjects) mixed design | Attitude towards vaccination against (dysomeria) Intention to get vaccinated against (dysomeria) | Perceived competence of denier's or advocate's message | Verbal aggressiveness Frequency of social media use | N = 310; U.S. adults; online sample (Prolific); $M_{Age}$ = 34.46, $SD_{Age}$ = 11.25; n = 155 women, n = 147 men, n = 8 non-binary participants; n = 17 high, n = 292 middle education | Social media discussion about vaccination against fictitious disease |
| 3 | 2 (hostile denier vs. neutral denier; between subjects) × 3 (hostile advocate vs. neutral advocate vs. advocate absent; between subjects) × 2 (time of measurement: before vs. after the debate; within subjects) mixed design | Attitude towards vaccination against (dysomeria) Intention to get vaccinated against (dysomeria) | Perceived competence of denier's or advocate's message | Verbal aggressiveness Need for cognition Issue involvement | N = 1200; U.S. adults; online sample (Prolific); $M_{Age}$ = 33, $SD_{Age}$ = 12.17, n = 585 women, n = 585 men, n = 28 non-binary participants; n = 47 high, n = 1141 middle, n = 7 low education | Social media discussion about vaccination against fictitious disease |
| 4 | 2 (hostile denier vs. neutral denier; between subjects) × 3 (hostile advocate vs. neutral advocate vs. advocate absent; between subjects) × 2 (time of measurement: before vs. after the debate; within subjects) mixed design | Attitude towards GM food in general Intention to buy GM variant of favourable food | Perceived competence of denier's or advocate's message | Verbal aggressiveness Need for cognition Issue involvement | N = 1195; U.S. adults; online sample (Prolific); $M_{Age}$ = 33.71, $SD_{Age}$ = 12.12, n = 610 women, n = 558 men, n = 25 non-binary participants; n = 51 high, n = 1131 middle, n = 8 low education | Fictitious social media discussion about GM food |

M denotes the mean value. SD denotes the standard deviation. High education means that participants reported having a doctorate/PhD degree. A medium level of education means that participants reported having a high school diploma or higher level of education but no doctorate/PhD degree. Some participants did not report information on age (Experiments 1 and 2: n = 2; Experiment 3: n = 2; Experiment 4: n = 3), gender (Experiment 1: n = 1; Experiments 1 and 2: n = 1; Experiments 3 and 4: n = 2), or education (Experiment 1: n = 3; Experiment 2: n = 1; Experiments 3 and 4: n = 5).

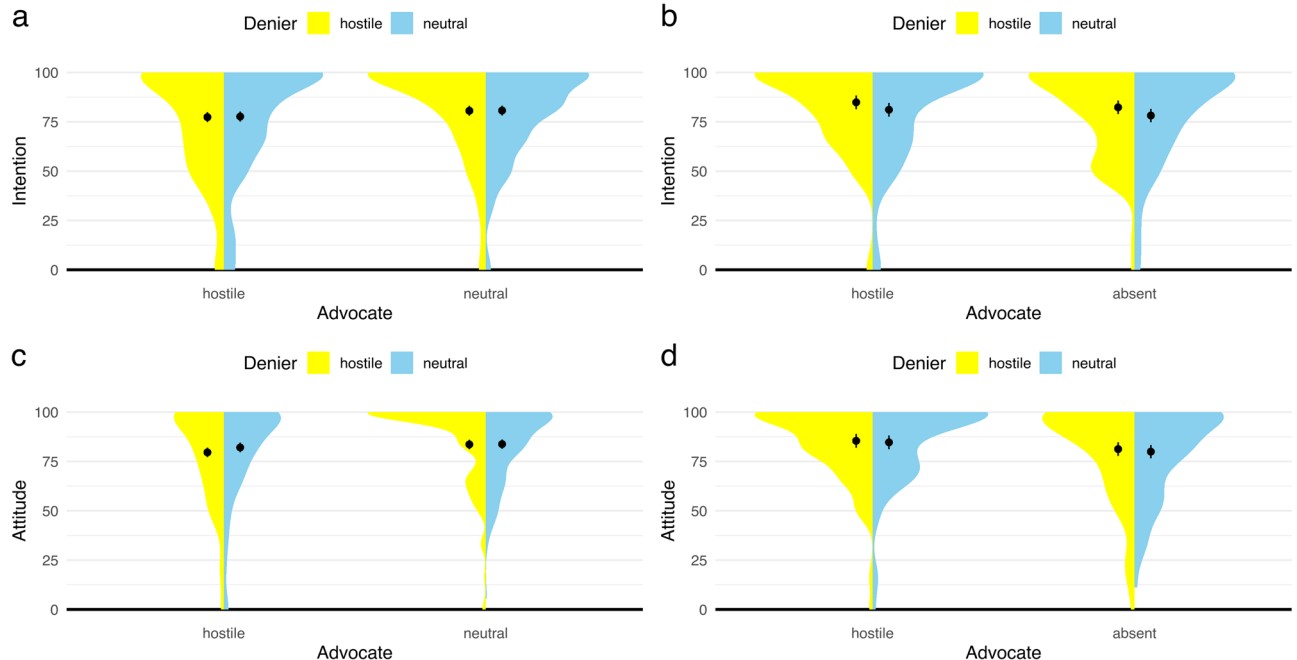

**Fig. 1 Individuals' attitude and intention scores after exposure to misinformation and rebuttal in Experiment 1 and Experiment 2.** The *y*-axes represent post scores for intention or attitude and the *x*-axes represent experimental conditions. Overall, the overlapping error bars indicate only small differences between experimental conditions on persuasiveness. Point estimates represent mean post scores of intentions in Experiment 1 (**a**: $n = 521$) and Experiment 2 (**b**: $n = 310$) or mean post scores of attitudes in Experiment 1 (**c**: $n = 521$) and Experiment 2 (**d**: $n = 310$). Mean post scores are adjusted for baseline values. Error bars represent 95% confidence intervals. Shaded areas indicate the density of observations.

packages for analyses: metafor version 3.8.1, emmeans 1.8.2, dplyr 1.0.10, pryr 0.1.5, Cairo 1.6.0, car 3.1.1, ggpubr 0.5.0, ggplot2 3.4.0, foreign 0.8.83, PROCESS for R version 4.0. We used SPSS version 28 for data provided in tables in the Supplementary Information. For equivalence testing, we used the TOSTER spreadsheet[52].

Following best practices, we analysed hypotheses about the direct effects of hostility with preregistered ANCOVA models[53]. That is, we compared post values of attitude and intention between conditions and controlled for baseline values (two-sided, significance threshold = 0.05). Tests to determine whether the data met the assumptions of the statistical tests are reported (Supplementary Table 6). Using variance ratio criteria, we found no evidence of a violation of the homogeneity of variances. Shapiro-Wilk tests were significant and indicated violations of normality. Given the robustness of ANOVA analyses against violations of normality[54], we proceeded with the preregistered analyses. In addition, when testing primary analyses with non-parametric tests, the results remained the same. All outcome variables in ANCOVA models are transformed into percentages of maximum possible scores (POMP)[55]. For instance, a difference of one point (20%) between neutral rebuttal and hostile rebuttal conditions on the original five-point attitude scale would correspond to a difference of 20 units (%) on the POMP scale. Since each outcome variable in the models has a range of 0-100 following the POMP transformation, using POMP values makes it simple to comprehend the model parameters. Visualisations of distributions and ANCOVA results for all single experiments are reported in Figs. 1 and 2.

We also analysed aggregated results of direct effects of hostility across all experiments using internal random effects meta-analyses. Effect sizes in meta-analyses are standardised mean differences. Mean differences are adjusted for baseline values and standardisers are pooled standard deviations for all of the cells in the respective design[56]. In addition, we used equivalence testing to test whether effect sizes are trivial (TOSTER spreadsheet[52]). That is, we explored whether the observed averaged effect sizes were significantly within the equivalence bounds of $d = 0.2$ and $d = -0.2$. The smallest effect size of interest (SESOI: $d = 0.2$) was preregistered (Experiment 3: https://aspredicted.org/7ui2w.pdf; Experiment 4: https://aspredicted.org/tv3sw.pdf) and chosen based on conventions for a small effect size[57]. This SESOI was considered meaningful for persuasion research, given the results of previous meta-analyses on the persuasive effectiveness of message design choices (median mean *r*s of about r = 0.10)[58]. In Experiment 1, we additionally tested whether the observed effects of the deniers and advocates on the attitudes or on the intentions were mediated by how much the advocate's responses were expected—in turn possibly moderated by the valence of the expectancy violation—via mediated moderation models using the PROCESS macro for R (model 14). For Experiments 3 and 4, we tested whether the effects of the denier or the advocate on the attitudes or intentions were mediated by the perceived competence of the respective comments using the PROCESS macro for R (model 4). Exploratively, we investigated the moderating role of the participants' verbal aggressiveness, need for cognition, issue involvement, and frequency of social media use for the observed effects using the PROCESS macro for R (model 1). All statistical tests that are based on symmetric, two-tailed distributions were two-sided. The only exemption are the TOST tests for equivalence. All reported confidence intervals are 95% intervals except for TOST tests for equivalence where 90% confidence intervals (one-sided tests) are used[52].

**Procedure and material.** The general procedure was similar in all experiments. Participants were randomly assigned to one of four conditions (Experiments 1 and 2) or six conditions (Experiments 3 and 4). An automatic randomisation mechanism provided by the Enterprise Feedback Suite (EFS) by Questback was used for

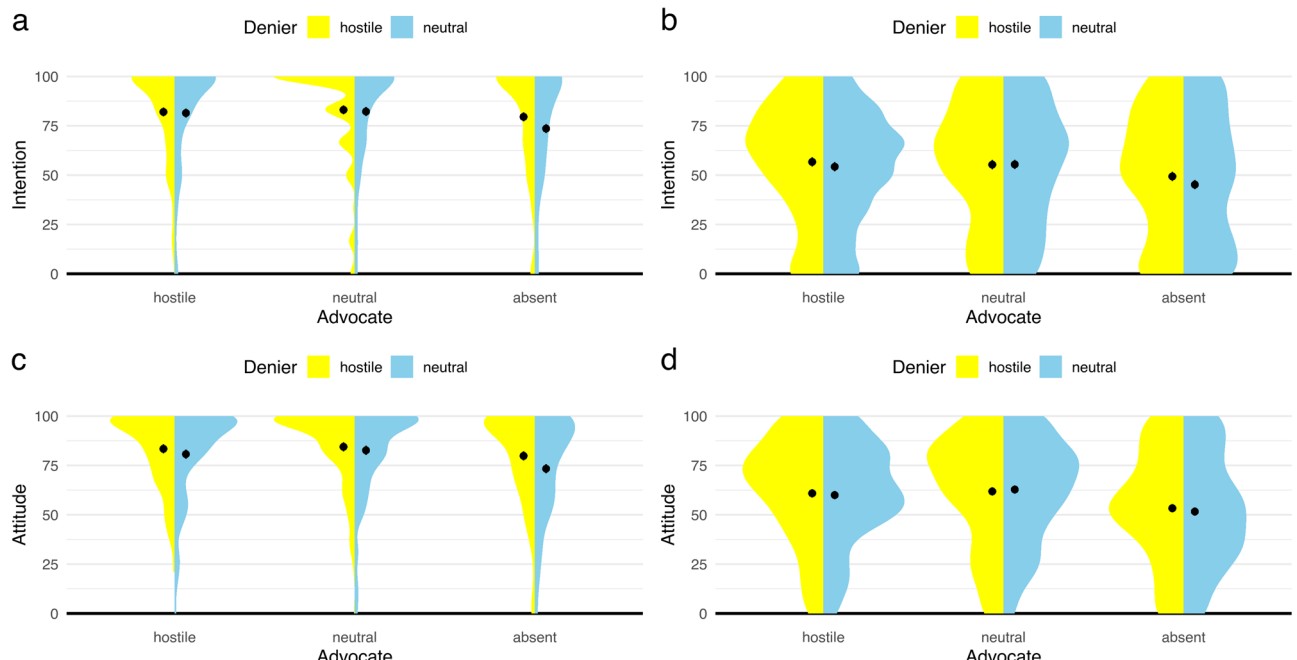

**Fig. 2 Individuals' attitude and intention scores after exposure to misinformation and rebuttal in Experiment 3 and Experiment 4.** The *y*-axes represent post scores for intention or attitude and the *x*-axes represent experimental conditions. Overall, the overlapping error bars indicate only small differences between hostile and neutral conditions on persuasiveness. Differences between rebuttal and rebuttal-absent conditions across experiments indicate a consistent benefit of using rebuttal. Point estimates represent mean post scores of intentions in Experiment 3 (**a**: *n* = 1200) and Experiment 4 (**b**: *n* = 1195) or mean post scores of attitudes in Experiment 3 (**c**: *n* = 1200) and Experiment 4 (**d**: *n* = 1195). Mean post scores are adjusted for baseline values. Error bars represent 95% confidence intervals. Shaded areas indicate the density of observations.

randomisation. At the start of the study, the software randomly selected which rebuttal information was to be communicated to participants. Owing to this randomisation procedure, the investigators were blind to the group allocation process. After consenting to participation, context information was given on the topic (Experiments 1–3: vaccine against the fictitious disease Dysomeria; Experiment 4: genetically modified foods). Participants then indicated their attitude (regarding perceived necessity, benefit, and reasonableness in Experiments 1 and 2, and additionally perceived safety in Experiments 3 and 4) towards the topic in question as well as the intention to perform the associated behaviour (Experiments 1–3: getting vaccinated; Experiment 4: buying GM foods); those variables were assessed once again together with further, potential moderating or mediating variables after the following manipulation.

As our manipulation, participants were prompted to read a fictitious conversation on a fictitious social media platform #YourVoice about the respective topic, wherein a science denier would reply to an innocuous post by spreading misinformation and an advocate for science would rebut the misinformation. Both discussants made two comments in a back-and-forth manner. The paradigm and messages from the denier and the rebuttals closely resembled those of previous research[9,42]. Depending on condition, swear words, direct insults and words in capital letters were either included (i.e., hostile message) in the messages or were left out (i.e., neutral message). The idea that the use of swear words and insults can be an indicator of hostility is based on the assumption that written text, like social media comments, reflects the cognitive and affective processes of the author. This assumption is widely shared across psychological and communication research and provides the basis for research on psycholinguistic dictionaries[59]. For example, textual analysis programs like Linguistic Inquiry and Word Count (LIWC) have been used to categorise whether public discussions are dominated

by hostile or civil language[59,60]. The message features of hostile language for this study were adopted from previous research[25,61,62]. An example is provided in Fig. 3. All misinformation and rebuttals are provided online[63]. At the end of all experiments, participants provided some demographics and were debriefed.

**Reporting summary**. Further information on research design is available in the Nature Portfolio Reporting Summary linked to this article.

## Results

**Manipulation checks.** In some studies, the protective effects of rebuttal may not be detected because the misinformation has no impact, that is, the science denier is not persuasive. To assess the potential impact of misinformation, we analysed pre-post change scores of primary dependent variables in the neutral-language conditions without rebuttal (i.e., rebuttal absent). Attitudes and intentions decreased after exposure to the science denier in all experiments with a rebuttal absent condition (Experiment 2: Attitude: $t = 4.89(80)$, $p < 0.001$, $d = 0.54$ [0.31, 0.78], Intention $t = 4.66(80)$, $p < 0.001$, Cohen's $d = 0.52$ [0.28, 0.75]; Experiment 3: Attitude: $t = 7.43(200)$, $p < 0.001$, $d = 0.52$ [0.38, 0.67], Intention: $t = 7.87(200)$, $p < 0.001$, $d = 0.56$ [0.41, 0.70]; Experiment 4: Attitude: $t = 7.68(199)$, $p < 0.001$, $d = 0.54$ [0.39, 0.69], Intention: $t = 4.98(199)$, $p < 0.001$, $d = 0.35$ [0.21, 0.49]). Thus, the protective effects of rebuttal were expected to be detectable.

In addition, we analysed the perceived politeness of hostile versus neutral misinformation and hostile versus neutral rebuttal to check whether manipulations of hostility were perceived as different in tone by participants. As intended, results reveal that hostile messages were perceived as significantly less polite than neutral messages in all experiments (Experiment 1: Denier:

## Experimental Stimulus

| **Neutral** | **Hostile** |
|---|---|
| **Misinformation:** "The lack of safety is an important issue of the dysomeria vaccine. The side effects and risks of the vaccine are incalculable. As a patient, you do not know how the body reacts to the vaccine before administration. Even if you feel healthy immediately after the shot, harmful substances may have entered your body. Doctors cannot guarantee in advance that there will not be any complications. In my opinion, you cannot expect any fellow citizen to vaccinate as long as the vaccine is not 100% safe. Surely it is not too much to ask that a product that is injected into a healthy human body is 100% safe." | **Misinformation:** "Everybody should know, the safety of the dysomeria vaccine is a complete joke! The side effects and risks of the vaccine are INCALCULABLE, you lunatic!!! As a patient, you have no fucking clue how your body reacts to that crap before administration. Even if you FEEL healthy immediately after the shot, harmful substances may have entered your body. The fucking QUACKS cannot guarantee in advance that there will not be any complications. They cannot expect any fellow citizens to vaccinate as long as this shit is not 100% safe!! Surely, it is not too much to fucking ask that a product INJECTED INTO A HEALTHY HUMAN BODY be completely fucking safe!?!!" |
| **Rebuttal:** "You demand 100% safety from the vaccine against dysomeria. In science, this argument is called *impossible expectation*. It is an impossible expectation because science can never guarantee 100% safety for any medical product, neither for aspirin nor for heart surgery. Any treatment poses a residual risk of complications for patients either during or after treatment. The scientific evidence is clear: The vaccine against dysomeria is a safe way to avoid the disease. The risk of dysomeria by far exceeds the risk from vaccination. This is why it is recommended for citizens of all ages to vaccinate against the DS virus. And please let me add the following regarding the safety of the vaccine: Health experts follow a very strict protocol to ensure the high quality of vaccines in the Federal States. This also is demonstrated by the fact that every batch of the vaccine against dysomeria is constantly monitored and independently screened by official control laboratories." | **Rebuttal:** "Do you seriously demand complete fucking safety from the vaccine against dysomeria?! In SCIENCE, this sort of idiotic argument is called *impossible expectation*. It is a fucking impossible expectation because science can never guarantee 100% safety for any medical product, neither for aspirin nor for heart surgery. I mean, for fuck's sake, any treatment poses a residual risk of complications for patients either during or after treatment. Do some research, you piece of shit: the scientific evidence is CLEAR!! The vaccine against dysomeria is a safe way to avoid the fucking disease. The risk of dysomeria by far exceeds the risk from vaccination!!! This is why it is recommended for citizens of all ages to vaccinate against the DS virus. And for my fucking sanity's sake, let me add the following regarding the safety of the vaccine: Health experts follow a very strict protocol to ensure the high quality of vaccines in the Federal States. This also is demonstrated by the fact that EVERY BATCH of the vaccine against dysomeria is constantly MONITORED AND INDEPENDENTLY SCREENED by official control laboratories. Or are you too stupid to understand that?" |

**Fig. 3 Excerpts from the experimental stimuli for rebutting science denialism in hostile social media discussions about vaccination.** Neutral misinformation and neutral rebuttal are adapted from previous research[9,42]. The adapted hostile versions include swear words, direct insults and words in capital letters. The displayed texts represent the four key combinations of misinformation (either neutral or hostile) and rebuttal (either neutral or hostile) as used in Experiments 1–3. The content was adapted to GM foods in Experiment 4. Full dialogues for all experiments are provided online[63].

$t = 23.12(519)$, $p < 0.001$, $d = 2.03$ [1.81, 2.24], Advocate: $t = 23.24(519)$, $p < 0.001$, $d = 2.04$ [1.82, 2.25]; Experiment 2: Denier $t = 19.26(308)$, $p < 0.001$, $d = 2.19$ [1.91, 2.47]; Experiment 3: Denier: $t = 30.69(1198)$, $p < 0.001$, $d = 1.77$ [1.64, 1.91], Advocate: $t = 26.27(798)$, $p < 0.001$, $d = 1.86$ [1.69, 2.02]; Experiment 4: Denier: $t = 37.10(1193)$, $p < 0.001$, $d = 2.15$ [2.00, 2.29]; Advocate: $t = 28.29(793)$, $p < 0.001$, $d = 2.01$ [1.84, 2.18]). Thus, persuasive effects of hostility were expected to be detectable.

**Experiment 1.** In this experiment, we analysed the persuasive impact of hostile deniers and hostile advocates compared to discussants using neutral language in a fictitious social media discussion about vaccination. Moreover, we analysed the indirect effects of hostility on the persuasiveness of the discussants through violations of the audience's expectancies and through the audience's competence judgements. Lastly, we explored whether individuals' verbal aggressiveness or frequency of social media use moderates the impact of hostility on the persuasiveness of misinformation and rebuttal.

First, the preregistered ANCOVA models revealed no statistically significant evidence that the tone of the misinformation influenced the audience's intention to get vaccinated or attitude towards vaccination (intention: $F(1, 516) = 0.02$, $p = 0.884$; attitude: $F(1, 516) = 1.16$, $p = 0.281$). In fact, differences between estimated means of hostile and neutral misinformation conditions were neglectable for both outcome measures (intention: $Mean_{difference} = -0.19$, $d = -0.01$ [-0.18, 0.16]; attitude: $M_{diff} = -1.33$, $d = -0.09$ [-0.27, 0.08]; Fig. 4a, b).

In contrast, ANCOVA models confirmed that the use of hostile language by the advocate for science damaged the effectiveness of the rebuttal. That is, individuals who read neutral rebuttals indicated higher attitudes towards vaccination and higher intentions to get vaccinated compared to individuals who read hostile rebuttals (attitude: $F(1, 516) = 5.31$, $p = 0.022$; intention: $F(1, 516) = 5.88$, $p = 0.016$). However, differences between means indicated only conventionally small effect sizes (intention: $M_{diff} = -3.13$, $d = -0.21$ [-0.39, -0.04]; attitude: $M_{diff} = -2.84$, $d = -0.20$ [-0.38, -0.03]; Fig. 4c, d). We did not find any evidence for interaction effects in the ANCOVA models (Supplementary Table 7).

Preregistered mediation models revealed no statistically significant evidence that expectancy violations explain the effect of hostility on the persuasiveness of rebuttal (Supplementary Table 8). In fact, we found no statistically significant evidence that rebuttals that used the same tone as the misinformation (i.e., hostile rebuttal following hostile misinformation and neutral rebuttal following neutral misinformation) were more expected by the audience than rebuttals that differed in tone from the misinformation (i.e., hostile rebuttal following neutral misinformation and neutral rebuttal following hostile misinformation; attitude: $B = 2.10$ [-1.91, 6.19], $p = 0.322$; intention: $B = 2.02$ [-2.05, 6.14], $p = 0.346$). Moreover, we found no statistically significant evidence that individuals' attitudes towards vaccination or intentions to get vaccinated were influenced by expectancy ratings (attitude: $B = -0.08$ [-0.12, 0.28], $p = 0.308$; intention: $B = 0.03$ [-0.13, 0.21], $p = 0.668$).

Additional exploratory mediation models revealed indirect effects of hostility of rebuttal on individuals' attitude towards

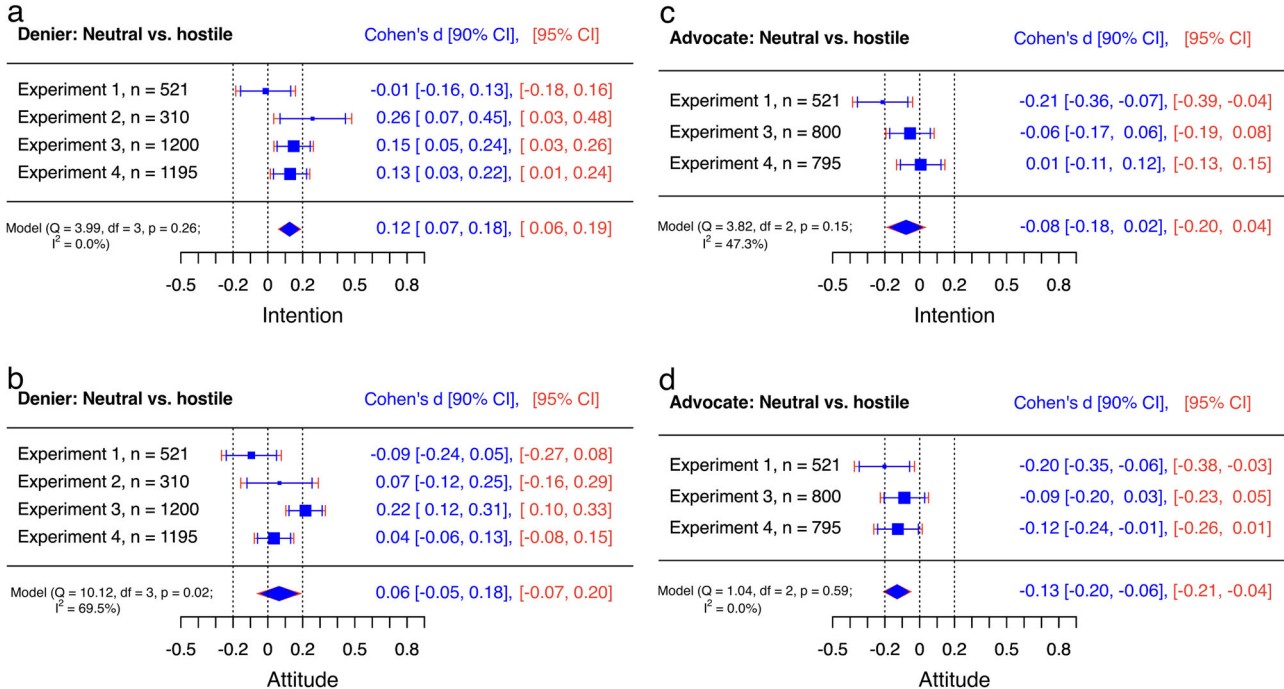

**Fig. 4 Trivial to small effects of hostility on the persuasiveness of misinformation and rebuttal.** Internal meta-analyses of neutral misinformation versus hostile misinformation (**a**, **b**) and neutral rebuttal versus hostile rebuttal (**c**, **d**) on intention and attitude of the audience using random effects models. Negative effect sizes indicate that the audience's attitude towards a behaviour dismissed by science deniers (**b**, **d**) or the intention to perform this behaviour (**a**, **c**) was lower when the respective speaker used hostile compared to neutral language. Thus, for the advocate (**c**, **d**), negative effect sizes indicate a decrease in the persuasive power of rebuttal when hostile language was used, while for the denier (**a**, **b**), positive values indicate a decrease in the persuasive power of misinformation when hostile language was used. The y-axes represent experiments, and the x-axes represent Cohen's ds. Mean differences are adjusted for baseline values and standardisers are pooled standard deviations for all of the cells in the respective design[56]. Diamonds show averaged effects. Red error bars indicate 95% confidence intervals. Blue error bars indicate 90% confidence intervals. The dotted central line is the zero line. If the red confidence interval does not include the zero line, then the effect is significantly different from zero. The left (−0.2) and right (0.2) dotted lines mark the equivalence bounds. If the effect lies between these lines and the blue confidence interval does not include any of the equivalence bounds, then the result is significantly equivalent, i.e., trivial. Q tests and $I^2$ indicate heterogeneity of results.

vaccination and intentions to get vaccinated through the audience's perceived competence of the discussant (Model 1 attitude: indirect effect $\beta = 4.15$ [1.90, 7.05]; Model 2 intention: $\beta = 3.30$ [1.44, 5.59]). Indirect effects were also found for the hostility of misinformation about vaccination (Model 3 attitude: $\beta = −3.60$ [−6.02, −1.59]; Model 4 intention: $\beta = −3.82$ [−6.45, −1.78]; Supplementary Tables 9 and 10). More specifically, a neutral rebuttal was perceived as a more competent contribution than a hostile rebuttal (Model 1: $B = 30.78$ [16.94, 44.22], $p = < 0.001$; Model 2: $B = 30.38$ [16.80, 43.73], $p < 0.001$) and neutral misinformation was perceived as a more competent contribution than hostile misinformation (Model 3: $B = 33.52$ [17.97, 49.52], $p = < 0.001$; Model 4: $B = 35.27$ [19.96, 50.98], $p < 0.001$). Increasing competence judgements of rebuttal, in turn, predicted more positive attitudes towards vaccination and higher intentions to get vaccinated among the audience (Model 1: $B = 0.13$ [0.08, 0.20], $p < 0.001$; Model 2: $B = 0.11$ [0.06, 0.17], $p < 0.001$) and increasing competence judgements of misinformation about vaccination predicted more negative attitudes and lower intentions among the audience (Model 3: $B = −0.11$ [−0.16, −0.06], $p < 0.001$; Model 4: $B = −0.11$ [−0.16, −0.06], $p < 0.001$).

Lastly, explorative analyses revealed that the (non-)effectiveness of hostility of the denier was a function of the audiences' social media use. Individuals with a high frequency of using social media reported higher positive attitudes towards vaccination after reading a hostile misinformation about vaccination compared to a neutral misinformation. This effect was absent for individuals with a low frequency of social media use. However, this

moderator effect was not significant for the intention to get vaccinated and was not present for the advocate for science (Supplementary Tables 11 and 12). Moreover, there was no statistically significant evidence that individuals' verbal aggressiveness moderated the effect of hostility on attitude and intention (Supplementary Tables 13 and 14).

**Experiment 2**. Experiment 2 was identical to Experiment 1 with two exceptions. We replaced the neutral advocate condition with an advocate absent condition to test if hostile rebuttals are more effective than leaving the stage to the denier in social media discussions about vaccination. Moreover, we dropped the hypothesis that expectancy violations may explain the effects of hostility.

In this experiment, the tone of misinformation played a role for the persuasiveness of the denier. Individuals who read neutral misinformation about vaccination indicated lower intentions to get vaccinated compared to individuals who read hostile misinformation, $F(1, 305) = 5.13$, $p = 0.024$. However, this effect was not observed for attitude towards vaccination, $F(1, 305) = 0.35$, $p = 0.556$, and again differences between estimated means of hostile and neutral conditions were small to neglectable (intention: $M_{diff} = 3.97$, $d = 0.26$ [0.03, 0.48]; attitude: $M_{diff} = 1.04$, $d = 0.07$ [−0.16, 0.29]; Fig. 4a, b).

Next, the persuasiveness of hostile rebuttal was compared to the advocate being absent from the social media discussion. Individuals who read hostile rebuttals had more positive attitudes towards vaccination compared to individuals who read no

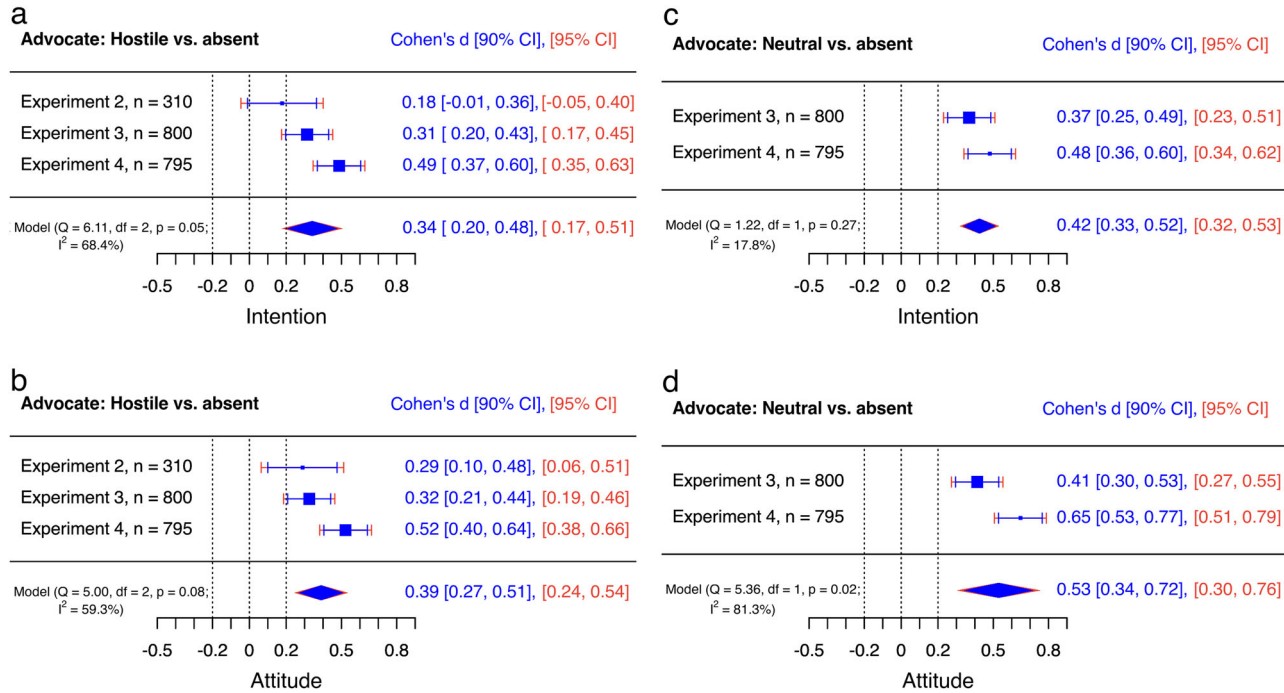

**Fig. 5 Small to medium effects of hostile and neutral rebuttal compared to no response.** Internal meta-analyses of hostile rebuttal (**a, b**) and neutral rebuttal (**c, d**) versus no response on intention and attitude of the audience using random effects models. Positive effect sizes indicate that the audience's attitude towards a behaviour dismissed by science deniers (**b, d**) or the intention to perform this behaviour (**a, c**) was higher when the advocate used hostile (**a, b**) or neutral (**c, d**) rebuttal compared to no response. The y-axes represent experiments, and the x-axes represent Cohen's *d*s. Mean differences are adjusted for baseline values and standardisers are pooled standard deviations for all of the cells in the respective design[56]. Diamonds show averaged effects. Red error bars indicate 95% confidence intervals. Blue error bars indicate 90% confidence intervals. The dotted central line is the zero line. If the red confidence interval does not include the zero line, then the effect is significantly different from zero. The left (−0.2) and right (0.2) dotted lines mark the equivalence bounds. If the effect lies between these lines and the blue confidence interval does not include any of the equivalence bounds, then the result is significantly equivalent, i.e., trivial. Q tests and $I^2$ indicate heterogeneity of results.

rebuttal, but we found no statistically significant evidence for a difference between the hostile and no rebuttal conditions on individuals' intentions to get vaccinated (attitude: $F(1, 305) = 6.43$, $p = 0.012$, intention: $F(1, 305) = 2.43$, $p = 0.119$). Differences between estimated means of hostile and no rebuttal conditions were small (intention: $M_{diff} = 2.73$, $d = 0.18$ [−0.05, 0.40]; attitude: $M_{diff} = 4.45$, $d = 0.29$ [0.06, 0.51]; Fig. 5a, b). We did not find any statistically significant evidence for interaction effects in the ANCOVA models (Supplementary Table 15).

Similar to Experiment 1, additional exploratory mediation models revealed indirect effects of hostility of misinformation on individuals' attitudes towards vaccination and intentions to get vaccinated through the audience's competence judgement of the denier (Model 1 attitude: $ß = −4.72$ [−8.15, −2.10]; Model 2 intention: $ß = −4.75$ [−8.61, −1.78]; Supplementary Table 16). Again, neutral misinformation about vaccination was perceived as a more competent contribution than hostile misinformation (Model 1: $B = 29.14$ [15.89, 42.78], $p = < 0.001$; Model 2: $B = 28.36$ [14.59, 42.17], $p < 0.001$) and increasing competence judgements of misinformation predicted lower attitudes towards vaccination and lower intentions to get vaccinated among the audience (Model 1: $B = −0.16$ [−0.24, −0.09], $p < 0.001$; Model 2: $B = −0.17$ [−0.25, −0.09], $p < 0.001$). An analysis of the indirect effect of hostility on rebuttal was not possible because neutral rebuttal was not tested in this experiment.

Again, explorative analyses revealed no statistically significant evidence that individuals' verbal aggressiveness or frequency of social media use moderated the effect of hostility on attitudes towards vaccination and intentions to get vaccinated (Supplementary Tables 17 and 18).

**Experiment 3**. In this experiment, we replicated the results of Experiment 1 and Experiment 2 in a single study by testing all previous experimental conditions in a 2 (hostile denier versus neutral denier; between subjects) × 3 (hostile advocate versus neutral advocate versus advocate absent; between subjects) × 2 (time of measurement: before versus after the debate; within subjects) mixed design. Furthermore, we preregistered the indirect effects of hostility on persuasiveness through competence judgements and explored the need for cognition and issue involvement as additional potential moderators of the (non-) effectiveness of hostility in the fictitious social media discussion about vaccination.

Individuals reported lower intentions to get vaccinated and lower positive attitudes towards vaccination when reading a neutral misinformation about vaccination compared to a hostile one (intention: $F(1, 1193) = 6.59$, $p = 0.010$; attitude: $F(1, 1193) = 14.10$, $p < 0.001$). Again, differences between estimated means indicated only small effect sizes (intention: $M_{diff} = 2.44$, $d = 0.15$ [0.03, 0.26]; attitude: $M_{diff} = 3.66$, $d = 0.22$ [0.10, 0.33]; Fig. 4a, b). We did find evidence for an interaction effect of experimental factors on individuals' intentions to get vaccinated but not on individuals' attitudes towards vaccination. That is, individuals reported lower intentions to get vaccinated when reading a neutral misinformation compared to a hostile one only when rebuttal was absent (Supplementary Table 19).

ANCOVA models also revealed significant differences between rebuttal conditions (attitude: $F(2, 1193) = 18.88$, $p < 0.001$; intention: $F(2, 1193) = 15.85$, $p < 0.001$). Planned contrasts indicated that neutral rebuttal was more effective in mitigating the impact of the denier compared to the advocate being absent in

the social media discussion (intention: $M_{diff} = 6.09$, $t = 5.22(1193)$, $p < 0.001$, $d = 0.37$ [0.23, 0.51]; attitude: $M_{diff} = 6.97$, $t = 5.83(1193)$, $p < 0.001$, $d = 0.41$ [0.27, 0.55]; Fig. 5c, d). The same benefit was found for the hostile rebuttal compared to the advocate being absent (intention: $M_{diff} = 5.17$, $t = 4.44(1193)$, $p < 0.001$, $d = 0.31$ [0.17, 0.45]; attitude: $M_{diff} = 5.49$, $t = 4.60(1193)$, $p < 0.001$, $d = 0.32$ [0.19, 0.46]; Fig. 5a, b). In addition, the direct comparison revealed no significant difference between hostile and neutral rebuttals on individuals' intention to get vaccinated or attitude towards vaccination (intention: $M_{diff} = -0.92$, $t = -0.79(1193)$, $p = 0.712$, $d = -0.06$ [−0.19, 0.08]; attitude: $M_{diff} = -1.48$, $t = -1.24(1193)$, $p = 0.430$, $d = -0.09$ [−0.23, 0.05]; Fig. 4c, d).

Additional confirmatory mediation models revealed indirect effects of hostility of rebuttal (Model 1 attitude: $\beta = 2.10$ [0.73, 3.87]; Model 2 intention: $\beta = 1.90$ [0.68, 3.52]) and misinformation about vaccination (Model 3 attitude: $\beta = -4.74$ [−6.13, −3.41]; Model 4 intention: $\beta = -3.37$ [−4.69, −2.15]) on individuals attitude towards vaccination and intentions to get vaccinated through the audience's perceived competence of the discussants (Supplementary Tables 20 and 21). Again, neutral rebuttal was perceived as a more competent contribution than hostile rebuttal in the social media discussion (Model 1: $B = 22.91$ [9.28, 36.35], $p < 0.001$; Model 2: $B = 24.49$ [10.59, 38.11], $p < 0.001$) and neutral misinformation about vaccination was perceived as a more competent contribution than hostile misinformation (Model 3: $B = 28.82$ [23.73, 33.74], $p < 0.001$; Model 4: $B = 28.90$ [23.88, 33.53], $p < 0.001$). Again, increasing competence judgements of rebuttal predicted more positive attitudes towards vaccination and higher intentions to get vaccinated among the audience (Model 1: $B = 0.09$ [0.05, 0.14], $p < 0.001$; Model 2: $B = 0.08$ [0.04, 0.12], $p < 0.001$) and increasing competence judgements of misinformation about vaccination predicted more negative attitudes towards vaccination and lower intentions to get vaccinated among the audience (Model 3: $B = -0.16$ [−0.21, −0.12], $p < 0.001$; Model 4: $B = -0.12$ [−0.16, −0.08], $p < 0.001$).

Explorative analyses revealed no statistically significant evidence that individuals' verbal aggressiveness or need for cognition or issue involvement moderated the effect of hostility on attitude and intention (Supplementary Tables 22–27).

**Experiment 4.** In the fourth experiment, we used the full design from Experiment 3 but changed the topic of the fictitious social media discussion to increase the generalisability of findings. That is, in Experiment 4, we focused on the safety of genetically modified foods.

Individuals reported lower intentions to buy GM food when reading a neutral misinformation about GM food compared to a hostile one (intention: $F(1, 1188) = 4.86$, $p = 0.028$). However, there was no statistically significant evidence of this effect for individuals' attitude towards GM food (attitude: $F(1, 1188) = 0.38$, $p = 0.538$), and differences between estimated means were small (intention: $M_{diff} = 2.16$, $d = 0.13$ [0.01, 0.24]; attitude: $M_{diff} = 0.54$, $d = 0.04$ [−0.08, 0.15]; Fig. 4a, b).

ANCOVA models revealed significant differences between rebuttal conditions on individuals' evaluation of GM food (intention: $F(2, 1188) = 31.05$, $p < 0.001$; attitude: $F(1, 1188) = 47.06$, $p < 0.001$). In line with Experiment 3, planned contrasts indicated that neutral rebuttal was more effective in mitigating the impact of the denier compared to the advocate being absent (intention: $M_{diff} = 8.12$, $t = 6.79(1188)$, $p < 0.001$, $d = 0.48$ [0.34, 0.62]; attitude: $M_{diff} = 9.84$, $t = 9.14(1188)$, $p < 0.001$, $d = 0.65$ [0.51, 0.79]; Fig. 5c, d). The same benefit over no rebuttal was found for hostile rebuttal in the social media

discussion (intention: $M_{diff} = 8.22$, $t = 6.86(1188)$, $p = 0.001$, $d = 0.49$ [0.35, 0.63]; attitude: $M_{diff} = 7.95$, $t = 7.37(1188)$, $p < 0.001$, $d = 0.52$ [0.38, 0.66]). Again, a direct comparison of rebuttals revealed no significant difference between hostile and neutral rebuttals on individuals' intentions to buy GM food or attitude towards GM foods (intention: $M_{diff} = 0.10$, $t = 0.09(1188)$, $p = 0.996$, $d = 0.01$ [−0.13, 0.15]; attitude: $M_{diff} = -1.89$, $t = -1.75(1188)$, $p = 0.186$, $d = -0.12$ [−0.26, 0.01]). We did not find evidence for an interaction effect of conditions (Supplementary Table 28).

In line with all previous experiments, confirmatory mediation models revealed indirect effects of hostility of rebuttal (Model 1 attitude: $\beta = 3.21$ [1.42, 5.47]; Model 2 intention: $\beta = 3.62$ [1.59, 5.93]) and misinformation (Model 3 attitude: $\beta = -3.44$ [−5.00, −2.04]; Model 4 intention: $\beta = -4.27$ [−5.90, −2.72]) on individuals attitude judgements and intentions through audience's perceived competence of the discussants (Supplementary Tables 29 and 30). Again, a neutral rebuttal was perceived as a more competent contribution than hostile rebuttal in the social media discussion (Model 1: $B = 25.47$ [12.64, 38.12], $p = < 0.001$; Model 2: $B = 24.54$ [11.52, 37.34], $p = < 0.001$) and neutral misinformation about GM food was perceived as a more competent contribution than hostile misinformation (Model 3: $B = 38.17$ [33.08, 43.03], $p = < 0.001$; Model 4: $B = 38.33$ [33.14, 43.37], $p = < 0.001$). Increasing competence judgements of rebuttal predicted more positive attitudes towards GM food and higher intentions to buy GM food among the audience (Model 1: $B = 0.13$ [0.09, 0.17], $p < 0.001$; Model 2: $B = 0.15$ [0.11, 0.19], $p < 0.001$) and increasing competence judgements of misinformation about GM food predicted more negative attitudes towards GM food and lower intentions to buy GM food among the audience (Model 3: $B = -0.09$ [−0.13, −0.05], $p < 0.001$; Model 4: $B = -0.11$ [−0.15, −0.07], $p < 0.001$).

Again, explorative analyses revealed no statistically significant evidence that individuals' verbal aggressiveness or need for cognition or issue involvement moderated the effect of hostility on attitude and intention (Supplementary Tables 31–36).

**Meta-analytic summary.** Figures 4–6 show meta-analytic results of all direct effects of hostility across the four experiments. Results revealed that individuals reported higher intentions to perform behaviours dismissed by science deniers when reading hostile misinformation compared to neutral misinformation, $d = 0.12$ [0.06, 0.19] (Fig. 4a). However, the averaged effect of hostility was not statistically significant for individuals' attitudes towards behaviours dismissed by science deniers, $d = 0.06$ [−0.07, 0.20], (Fig. 4b). Moreover, equivalence tests revealed trivially small effects of hostility for the denier: the blue confidence intervals in Fig. 4a, b reveal that both averaged effect sizes were significantly within the equivalence bounds of $d = 0.2$ and $d = -0.2$ (intention: $Z = -2.12$, $p = 0.017$; attitude: $Z = -1.99$, $p = 0.023$).

Rebutting misinformation consistently mitigated its impact: individuals reported higher attitudes towards behaviours dismissed by science deniers and higher intentions to perform these behaviours when reading a neutral rebuttal from an advocate for science compared to the advocate being absent (attitude: $d = 0.53$ [0.30, 0.76]; intention: $d = 0.42$ [0.32, 0.53]; Fig. 5c, d). The same pattern was found for hostile rebuttal (attitude: $d = 0.39$ [0.24, 0.54]; intention: $d = 0.34$ [0.17, 0.51]; Fig. 5a, b). The direct comparison of the rebuttal conditions shows that individuals who read neutral rebuttals indicated higher attitudes towards behaviours dismissed by science deniers compared to individuals who read hostile rebuttals, $d = -0.13$ [−0.21, −0.04]. However, the effect of hostility on the persuasiveness of the advocate for science was rather small and the effect of hostility was not statistically

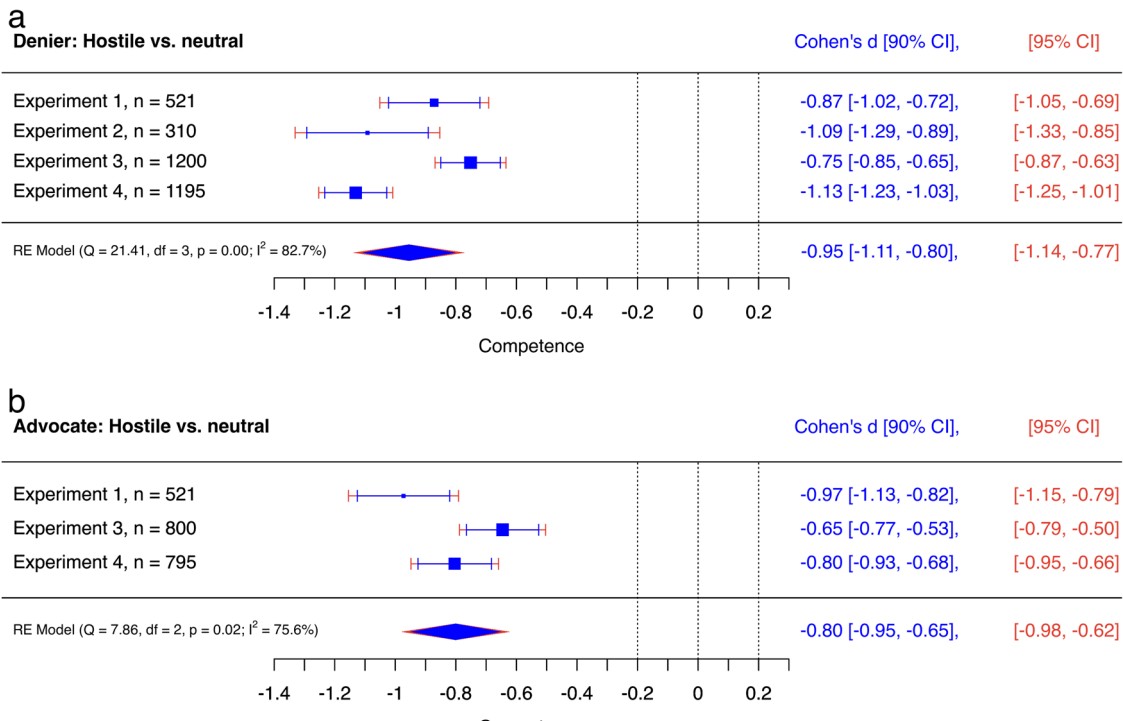

**Fig. 6 Medium to large effects of hostility on perceived competence of hostile discussants.** Internal meta-analyses of neutral misinformation versus hostile misinformation (**a**) and neutral rebuttal versus hostile rebuttal (**b**) on the audiences' perceived competence. Negative effect sizes indicate that the perceived competence of the denier's (**a**) or the advocate's (**b**) contribution was lower when the respective speaker used hostile compared to neutral language. The $y$-axes represent experiments, and the $x$-axis represent Cohen's $d$s. Diamonds show averaged effects. Red error bars indicate 95% confidence intervals. Blue error bars indicate 90% confidence intervals. The dotted central line is the zero line. If the red confidence interval does not include the zero line, then the effect is significantly different from zero. The left ($-0.2$) and right (0.2) dotted lines mark the equivalence bounds. If the effect lies between these lines and the blue confidence interval does not include any of the equivalence bounds, then the result is significantly equivalent, i.e., trivial. $Q$ tests and $I^2$ indicate heterogeneity of results.

significant for the second outcome measure (intention: $d = -0.08$ $[-0.20, 0.04]$; Fig. 4c, d). In fact, the blue confidence intervals in Fig. 4c reveal that the effect size for the intention was trivially small—significantly within the equivalence bounds of $d = 0.2$ and $d = -0.2$, $Z = 2.02$, $p = 0.022$.

While benefits for avoiding hostility were small to trivial on persuasive outcome measures, the damaging effects on perceived competence of the contribution were consistently strong for the denier, $d = -0.95$ $[-1.14, -0.77]$, as well as the advocate for science, $d = -0.80$ $[-0.98, -0.62]$, (Fig. 6). Thus, avoiding hostility may have limited effects on beliefs and behavioural intentions of a behaviour dismissed by science deniers but it strongly supports a positive image of the discussant among the audience.

**Robustness checks**. In Experiments 1 and 2, we compared participants' perceptions of the authenticity of hostile and neutral discussions to check whether hostile messages were perceived as less realistic (indicating that the manipulation was potentially confounded by authenticity). We found no statistically significant evidence of differences in these ratings between conditions (Supplementary Table 37).

In Experiments 3 and 4, we also explored the potential influence of social desirability on primary results. We excluded individuals who scored high on social desirability and repeated analyses of the direct effects of hostility on primary outcome measures as a robustness check (Supplementary Tables 38 and 39). The pattern of results did not differ.

In all experiments, we explored the potential influence of speeders on primary results. We excluded individuals that took less than 202 sec. (one standard deviation below the mean duration) to

complete the study and individuals that had no recorded duration due to interruptions (coded as $-1$ by the survey software). We repeated analyses of the direct effects of hostility on primary outcome measures without the excluded individuals ($N = 147$) as a robustness check (Supplementary Tables 40–43). The difference between hostile and neutral misinformation on individuals' intentions in Experiment 2 (Supplementary Table 41) and Experiment 4 (Supplementary Table 43) was not significant after excluding speeders. Moreover, the difference between hostile and no rebuttal on individuals' intentions in Experiment 2 (Supplementary Table 42) was significant after excluding speeders. Thus, we repeated all internal meta-analyses excluding speeders. The pattern of meta-analytic results, including equivalence tests, did not differ from the primary result reported in the main manuscript. Adjusted forests plots are provided online[63].

Lastly, in all experiments, we used a single-item attention check. In Experiments 1 and 2, participants were asked about the topic of the conversation they had been reading, whereas in Experiments 3 and 4, we integrated an item into the questionnaire, asking participants to select almost always true as their response. Between 96 and 99% of the samples passed the attention check. We repeated analyses of the direct effects of hostility on primary outcome measures without inattentive participants as a robustness check (Supplementary Tables 44–47). The pattern of results did not differ.

## Discussion
Across four preregistered experiments, we found that the use of hostile language plays a minor role in the effectiveness of rebuttals against misinformation about vaccination and GM foods in social

media discussions, at least on persuasive outcome measures such as attitudes and intentions of U.S. American audiences. Figure 4c, d reveals that a neutral rebuttal was only more effective than a hostile rebuttal in two out of six single evaluations and in one out of two internal meta-analyses. Moreover, additional equivalence tests revealed that the averaged effect sizes were only small to trivial. Thus, if an advocate for science is more verbally aggressive in nature or is getting provoked to respond in a hostile way, not all is lost. On the contrary, the experiments revealed that hostile rebuttals of misinformation about vaccination or GM foods were more effective in mitigating the damage of science denialism compared to not responding at all. Across experiments, five out of six single evaluations revealed the benefit of using hostile rebuttal with conventionally small to medium effect sizes (Fig. 5a, b). Thus, rebuttals remain an effective countermeasure against misinformation even when the debate heats up. This knowledge is specifically helpful because science deniers and online trolls are known to provoke emotional responses[64,65]—a strategy that, if successful, seems to have little impact on the effectiveness of rebuttal, at least when it comes to intentions and attitudes about vaccination and GM foods in online debates with U.S. American audiences.

But the choice of hostile language comes with a price. All single experiments revealed that hostile language is perceived as less competent than neutral language when discussing vaccination or GM foods and analyses of individual experiments and across experiments revealed conventionally medium to large effect sizes for these differences (Fig. 6). Thus, hostility may play a minor role for the direct persuasiveness of rebuttal in social media discussions about vaccination or GM foods, but in line with previous research[66], it does impact how rebuttal is perceived by U.S. American audiences. This negative effect of hostility may have unintended consequences for the advocate because perceived competence is discussed as a key element of building trust between advocates for science and target audiences[27,67].

The results revealed a similar pattern for the role of hostility on the impact of messages of science denialism. Misinformation about vaccination and GM foods was more impactful when the science denier avoided hostility in only four out of eight single evaluations (Fig. 4a, b). Again, effect sizes for single evaluations and averaged effects across experiments were conventionally very small. In fact, equivalence tests for both internal meta-analyses revealed that the estimates of average effect sizes for hostility are significantly smaller than the smallest effect size of interest. Thus, hostility plays a negligible role in the potential damage of misinformation about vaccination and GM foods in social media discussions.

Next, we analysed whether the impact of hostility on the persuasiveness of messages can be explained by the audiences' perceptions of a hostile contribution. Mediator analyses from all experiments revealed that the use of hostility undermined the perceived competence of a discussant's message in a social media discussion and thereby damaged the discussant's persuasive power—at least indirectly. These indirect effects were observed for the science denier and the advocate. This suggests that the small persuasive advantage of neutral over hostile messages was because hostile messages are perceived as a less competent contribution by U.S. American audiences of a social media discussion about vaccination or GM foods. We did not find evidence for the hypothesis that expectancy violations can explain the effects of hostility. In fact, we did not find any differences in expectancy evaluations across conditions in Experiment 1. It may be the case that participants think of hostility as a violation of how one should communicate (i.e., injunctive norm) but not necessarily as a violation of how individuals actually communicate on social media (i.e., descriptive norm). In fact, hostility in social media environments may be even expected to some degree[41]. Another explanation is that the advocate for science was presented as a

regular social media user rather than an official health authority. Expectancy ratings may be different if a medical doctor or spokesperson of a health authority uses hostility.

Interestingly, mediation models revealed indirect effects of hostility on the persuasiveness of the advocate for science and the science denier in all experiments, while direct effects remained rare. This may be the result of other hidden indirect effects of hostility that neutralise the damaging effect of hostility. In fact, research suggests that the audience may perceive an aggressive speaker as more authentic than a neutral speaker because she or he does not mince words[35]. Moreover, following the emotions as arguments approach, hostile messages may be beneficial for persuasion because they convey anger and show the audience more clearly how outrageous the standpoint of the opponent is[68]. In addition, hostility may suggest to the audience that the speaker holds a dominant position[69] and furthermore, hostile messages may be more engaging and thereby increase attention towards the hostile message[70]. Thus, some findings challenge the idea that hostility is necessarily harmful to the hostile speaker and that hostility can even be a tool of persuasion. This may explain the general weak direct effects of hostility on the persuasiveness of rebuttal observed in this and other studies with similar patterns[71].

Another reason for general weak effects of hostility on persuasiveness may be that only specific audiences are sensible towards the tone of a message. However, the results of this study did not find any evidence that the effects of hostility in social media discussions about vaccination or GM foods are stronger among potentially sensible audiences, such as individuals low in verbal aggressiveness. In fact, none of the tested characteristics of the receiver moderated the effect of hostility.

In this study, we focus on the persuasiveness of hostile messages on individuals' attitudes and intentions to get vaccinated or to buy GM foods. However, hostility can have other damaging effects which need to be considered when evaluating the potential dangers of hostility. Hostile rebuttals may be effective, but research suggests that they have the potential to increase polarisation among audiences that may have had less extreme views before the public discussion[72]. Moreover, when hostility is used frequently in public discussions about science, then this may foster toxic environments where blaming and argumentum ad hominem attacks threaten democratic dialogues that are based on the idea of a free and respectful exchange of opinions[73]. Lastly, hostile messages can have severe effects on the target of the message. For example, scientists who were more targeted by personal attacks during the COVID-19 pandemic reported that this experience had affected their readiness to speak up as an advocate for science in public—a chilling effect that may weaken the cause of science advocacy in the future[74]. In fact, our results revealed that the impact of misinformation is strongest when the advocate for science stays absent from the social media discussion (Fig. 5). Thus if hostility intimidates scientists into silence, then the impact of science deniers is likely to be greatest, at least when it comes to intentions and attitudes about vaccination and GM foods in online debates with U.S. American audiences.

In this study, we focus on the impact of hostility on an audience and not on the science denier. Best practice guidance suggests refraining from hostility when trying to convince a denier in a peer-to-peer discussion[75,76]. Our results add to that by showing that not losing temper can also result in better overall evaluations by U.S. American audiences in social media discussions about vaccination or GM foods.

## Limitations
We analysed several potential moderators of the effectiveness of hostility on the individual level. However, all samples in this

study are U.S. samples. That is, we did not vary the cultural context or participants' country of origin, which may play an important role in how hostility is perceived in discussions. We did not measure actual behaviours dismissed by science deniers (e.g., vaccination uptake) but behavioural intentions. Behavioural intentions do not necessarily translate into actual behaviours[77]. Hostility can be defined in terms of its cognitive (e.g., cynical beliefs and distrust towards others), affective (e.g., anger and disgust) and behavioural components (e.g., aggressive physical behaviour and verbal aggression)[78,79]. We chose to focus on the verbal expression of hostility in this study because it is common in social media discussions about polarised scientific issues[13–15]. This focus also means that findings cannot be generalised to other expressions of hostility. For example, if actual physical violence or more subtle non-verbal face expressions of anger and disgust are part of a persuasive episode, then this may impact the persuasiveness of individuals to a different extent than the use of verbal cues. The participants of the study were asked to make judgements about vaccination and GM food in fictitious social media scenarios. In real social media environments, individuals may judge hostile content in a discussion differently due to existing conversational norms for specific social media platforms or due to individuals' social relationships with discussants (e.g., the attacked person is perceived as a friend). Moreover, many individuals do not only observe social media content but interact and respond to it. In that case, the audience can quickly become part of the discussion and thereby become a target of hostility itself. Varying norms on social media platforms, varying social relationships with discussants and varying perceptions of being the target of hostility add a level of complexity to social media interactions which was not the focus of this experimental study. In addition, social media interactions are only one form of public discussion. Generalisation of findings to other formats is thus limited. For example, effects may be stronger when accompanied by visual expressions of hostility in televised discussions. Finally, we varied the hostility of a message without changing the content of the arguments. Some people become highly emotional when reacting in a hostile manner and may be unable to make an argument that they would make in a more neutral state. However, research reveals that hostile discussants are no less likely than neutral discussants to use evidence in support of their position[73].

Historically, science denialism "had nothing to do with flaws in the science, and had everything to do with market fundamentalism, political commitments to free market politics, and hostility to government action in the marketplace"[80]. In view of this perspective, hostility is an essential part of the toolbox of science deniers to gain the upper hand in what has been described as information warfare[81]. The present study reveals that advocates for science can still be effective even if they lose their temper in emotional social media discussions about vaccination or GM foods in the U.S. and that the impact of misinformation should not be underestimated, regardless of how unprofessional a science denier's tone may seem to the scientific community. The present study, however, also reveals that it is crucial to refrain from hostility when the goal is to be perceived as a competent information source.

## Data availability

The datasets for all experiments and figures are available at Open Science Framework[63]. Figures and datasets for forest plots of analyses without speeders and full stimulus materials are also provided at the Open Science Framework. The DOI to access the materials and datasets is https://doi.org/10.17605/OSF.IO/HG2Y8. The permanent weblink is https://osf.io/hg2y8/.

## Code availability

All code for data analyses associated with the current submission is available at Open Science Framework[63]. The DOI to access the code is https://doi.org/10.17605/OSF.IO/HG2Y8. The permanent weblink is https://osf.io/hg2y8/.

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

## Acknowledgements

P.S. acknowledges support from the European Commission (Horizon 2020 grant agreement No. 964728 JITSUVAX). P.S. and B.W. acknowledge support from the Faculty of Economics, Law and Social Sciences and the Faculty of Education at the University of Erfurt. The funders had no role in study design, data collection and analysis, decision to publish or preparation of the manuscript.

## Author contributions

P.S. contributed to conceptualisation, data curation, formal analysis, investigation, methodology, project administration, resources, validation, visualisation, writing—original draft, and writing—review and editing. B.W. contributed to conceptualisation, methodology, project administration, resources, validation, and writing—review and editing.

## Funding

## Competing interests

The authors declare no competing interests.
