## [Peer Review File · Communications Psychology]

28th Aug 23

Dear Philipp,

Thank you for your patience during the peer-review process. Your manuscript titled "The role of hostility when rebutting science denialism in public discussions" has now been seen by 2 reviewers, and I include their comments at the end of this message. They find your work of interest, but raised some important points. We are interested in the possibility of publishing your study in *Communications Psychology*, but would like to consider your responses to these concerns and assess a revised manuscript before we make a final decision on publication.

We therefore invite you to revise and resubmit your manuscript, along with a point-by-point response to the reviewers. Please highlight all changes in the manuscript text file.

Editorially, we consider it important that the revised manuscript not only contains a transparent discussion of limitations (under the subheading "limitations" within the "Discussion" section), but also makes explicit the exact conditions that were tested throughout all sections of the manuscript, from the Abstract onward. Both reviewers highlight examples of study characteristics that are important to mention throughout (like the nature of the sample, the outcome measure, and the context of the experiment).

Please use the following link to submit your revised manuscript, point-by-point response to the referees' comments (which should be in a separate document to any cover letter) and the completed checklist:

[link redacted]

I have attached a template file that clarifies the formatting of our published Articles and we ask you to ensure that your manuscript complies with this formatting guidance. I also highlighted in the file some issues that absolutely must be addressed in revision.

You do not have to include Author contributions and other de-anonymizing items at this stage, but you are free to do so if you please.

We hope to receive your revised paper within 4-6 weeks; please let us know if you aren't able to submit it within this time so that we can discuss how best to proceed.

Please do not hesitate to contact me if you have any questions or would like to discuss these revisions further. We look forward to seeing the revised manuscript and thank you for the opportunity to review your work.

Best regards,

Marike

Marike Schiffer, PhD
Chief Editor
Communications Psychology

EDITORIAL POLICIES AND FORMATTING

Editorial Policy: [Policy requirements](https://www.nature.com/documents/nr-editorial-policy-checklist.pdf) (Download the link to your computer as a PDF.)

Furthermore, please align your manuscript with our format requirements, which are summarized on the following checklist:

[Communications Psychology formatting checklist](https://www.nature.com/documents/commspsychol-style-formatting-checklist-article-rr.pdf)

* **CODE AVAILABILITY:** All Communications Psychology manuscripts must include a section titled "Code Availability" at the end of the methods section. In the event of publication, we require that the custom analysis code supporting your conclusions is made available in a publicly accessible repository; at publication, we ask you to choose a repository that provides a DOI for the code; the link to the repository and the DOI will need to be included in the Code Availability statement. Publication as Supplementary Information will not suffice. We ask you to prepare code at this stage, to avoid delays later on in the process.

* **DATA AVAILABILITY:**

All Communications Psychology manuscripts must include a section titled "Data Availability" at the end of the Methods section or main text (if no Methods). More information on this policy, is available at <http://www.nature.com/authors/policies/data/data-availability-statements-data-citations.pdf>.

At a minimum the Data availability statement must explain how the data can be obtained and

whether there are any restrictions on data sharing. Communications Psychology strongly endorses open sharing of data. If you do make your data openly available, please include in the statement:

We recommend submitting the data to discipline-specific, community-recognized repositories, where possible and a list of recommended repositories is provided at <http://www.nature.com/sdata/policies/repositories>.

If a community resource is unavailable, data can be submitted to generalist repositories such as [figshare](https://figshare.com/) or [Dryad Digital Repository](http://datadryad.org/). Please provide a unique identifier for the data (for example a DOI or a permanent URL) in the data availability statement, if possible. If the repository does not provide identifiers, we encourage authors to supply the search terms that will return the data. For data that have been obtained from publicly available sources, please provide a URL and the specific data product name in the data availability statement. Data with a DOI should be further cited in the methods reference section.

REVIEWERS' EXPERTISE:

Both reviewers have research expertise in science communication/health communication and effects of language on communication

REVIEWERS' COMMENTS:

Reviewer #1 (Remarks to the Author):

The paper investigates the impact of hostility in public debates involving science deniers and advocates. The findings indicate that hostility has minimal effects on the persuasiveness of discussants, as even hostile rebuttals seem to effectively counter misinformation. However, it does undermine the perceived competence of the discussant, casting doubt on their credibility as a trustworthy information source. This insight holds important implications for defining best practices and policies for science communication and dissemination on social media.

The topic of the paper is interesting. The paper is well-written, with clear language, and employs an appropriate methodology. I have some suggestions to enhance its overall clarity, which are detailed below.

- Given that the experiments exclusively involve participants from the US, I would suggest clarifying throughout the article that the results can be extrapolated to the U.S. population but may not

necessarily apply to other countries or cultures.

- Since the experiments are conducted in a (fictitious) social media platform, it would be important to emphasize throughout the article, starting from the title and abstract, that the results are primarily applicable to social media contexts and may not necessarily extend to other settings.

- It would be beneficial to offer a more precise definition of "impact of misinformation" within the context of the paper.

- While the four experiments are explained in detail, the paper's flow can occasionally become less straightforward. To enhance clarity, I would recommend summarizing the settings and key aspects of the four experiments in a table or figure, to provide an organized overview of the experimental setups.

- I appreciate the discussion of the limitations and implications of the research in the Discussion section. However, I would also suggest addressing the limitations associated with the laboratory setting where the experiments were conducted. While the experiments investigate hostility within a fictitious social media environment, it is important to acknowledge that real-world interaction dynamics can be different and influenced by a variety of factors. For instance, in real settings, the public might have direct connections with the discussants and hostile comments might be directed at individual members of the public. Expanding the discussion to encompass these points would provide a more comprehensive understanding of the study's applicability and potential limitations.

Reviewer #2 (Remarks to the Author):

The paper is a well-structured report of four experiments approaching an interesting and relevant question: How are science deniers influenced by science-friendly arguments? The studies are well conducted, the paper is in really good shape. I like to see the paper in press and feel that it is a significant contribution to the field. However, I have a few comments that might be addressed in a revision:

- To me the most important aspect addressed in the paper is the role of language and communication on this question. I would also like to see specific measures discussed, like LIWC (Pennebaker).

Also hostility is manipulated by inappropriate words. To me, this is something different than hostile. Please explain in more depth why you opt for this manipulation (and report this in the limitations).

- The two foci "science denier" and "advocate for science" need explanation. Is an advocate for science a scientist. On the word level an symmetry between these two positions seem to be indicated. However, I would like to see a more thorough reflection for both positions. In doing so, I would like to see a short passage on insights on deniers as such (see Laundry, <http://www.asheleylandrum.com>).

- Counterarguing and Advocacy are two concepts that fit the theoretical underpinning of the studies. Please elaborate more on these.

- From vaccination to gen-manipulated food is a big step. I see that the aim is to generalize the findings of Exp 3. However, a more thorough discussion on the role of topic is needed.

Minors: Try to use a balanced wording (e.g. skip “fortunately” on page 2, line 23)

We really appreciate the time and effort all reviewers have spent evaluating the manuscript as well as the constructive feedback we received.

We describe all the changes, point by point, below. In each case, we have listed both the comments and our responses. Where appropriate and reasonable in length, we have also included relevant sections of the revised manuscript to illustrate the changes.

Reviewer #1 Comments:

Comment 1: The paper investigates the impact of hostility in public debates involving science deniers and advocates. The findings indicate that hostility has minimal effects on the persuasiveness of discussants, as even hostile rebuttals seem to effectively counter misinformation. However, it does undermine the perceived competence of the discussant, casting doubt on their credibility as a trustworthy information source. This insight holds important implications for defining best practices and policies for science communication and dissemination on social media.

The topic of the paper is interesting. The paper is well-written, with clear language, and employs an appropriate methodology. I have some suggestions to enhance its overall clarity, which are detailed below.

Response: We thank the reviewer for the positive evaluation and the very constructive feedback.

Comment 2: Given that the experiments exclusively involve participants from the US, I would suggest clarifying throughout the article that the results can be extrapolated to the U.S. population but may not necessarily apply to other countries or cultures.

Response: Following this comment, we mention the country of origin of the samples throughout the text. We also added this as an information in our discussion of findings and made this limitation stronger in the limitation section on pp. 31:

“We analysed several potential moderators of the effectiveness of hostility on the individual level. However, all samples in this study are U.S. samples. That is, we did not vary the cultural context or participants’ country of origin which may play an important role in how hostility is perceived in discussions.”

We also added a new Table (Table 1, page 11) following the reviewers comment 5. In this table the country of origin is part of the study description. We believe this clarifies the context and also the generalisability of findings.

Comment 3: Since the experiments are conducted in a (fictitious) social media platform, it would be important to emphasize throughout the article, starting from the title and abstract, that the results are primarily applicable to social media contexts and may not necessarily extend to other settings.

Response: Following this comment, we use the term social media discussion rather than public discussion throughout the text to specify the scenario more precisely. We emphasize this throughout the article, including the title and abstract. We also added this as a limitation to the limitations section (see response to Comment 6) and added the setting as a characteristic of the studies in the new Table 1.

Comment 4: It would be beneficial to offer a more precise definition of "impact of misinformation" within the context of the paper.

Response: Thank you for this suggestion. Following this comment, we now provide a more detailed description of the impact of misinformation as part of a more detailed sub-Section (Manipulation Checks) in the Results Section. This is the first sub-section in the Results section so we make this point clear to readers from the beginning page 13:

“In some studies, protective effects of rebuttal may not be detected because the misinformation has no impact, that is, the science denier is not persuasive. To assess the potential impact of misinformation, we analysed pre-post change scores of primary dependent variables in the neutral-language conditions without rebuttal (rebuttal absent). Attitudes and intentions decreased after exposure to the science denier in all experiments with a rebuttal absent condition (Experiment 2: Attitude: -9.26 [-12.63, -5.88], Intention -8.85 [-12.24, -5.45]; Experiment 3: Attitude -12.19 [-14.60, -9.78], Intention: -11.94 [-14.23, -9.65]; Experiment 4: Attitude: -8.48 [-10.63, -6.33] Intention: -6.92 [-9.34, -4.49]; Supplementary Figure 3–4). Thus, protective effects of rebuttal were expected to be detectable.”

Comment 5: While the four experiments are explained in detail, the paper's flow can occasionally become less straightforward. To enhance clarity, I would recommend summarizing the settings and key aspects of the four experiments in a table or figure, to provide an organized overview of the experimental setups.

Response: We thank the reviewer for this suggestion. We added the suggested table as Table 1 in the main manuscript on page 11. We think the table increases the flow of the paper.

Comment 6: I appreciate the discussion of the limitations and implications of the research in the Discussion section. However, I would also suggest addressing the limitations associated with the laboratory setting where the experiments were conducted. While the experiments investigate hostility within a fictitious social media environment, it is important to acknowledge that real-world interaction dynamics can be different and influenced by a variety of factors. For instance, in real settings, the public might have direct connections with the discussants and hostile comments might be directed at individual members of the public. Expanding the discussion to encompass these points would provide a more comprehensive understanding of the study's applicability and potential limitations.

Response: We thank the reviewer for this suggestion. We added the reviewer's thoughts as part of the following new paragraph to the limitations section on page 32:

“The participants of the study were asked to make judgements about vaccination and GM food in fictitious social media scenarios. In real social media environments, individuals may judge hostile content in a discussion differently due to existing conversational norms for specific social media platforms or due to individuals' social relationships with discussants (e.g., the attacked person is perceived as a friend). Moreover, many individuals do not only observe social media content but interact and respond to it. In that case, the audience can quickly become part of the discussion and thereby become a target of hostility itself. Varying norms on social media platforms, varying social relationships with discussants and varying perceptions of being the target of hostility add a level of complexity to social media

interactions which was not in the focus of this experimental study. In addition, social media interactions are only one form of public discussions. Generalisation of findings to other formats, for example, radio or television discussions, are thus limited.”

Reviewer #2 Comments:

Comment 1: The paper is a well-structured report of four experiments approaching an interesting and relevant question: How are science deniers influenced by science-friendly arguments? The studies are well conducted, the paper is in really good shape. I like to see the paper in press and feel that it is a significant contribution to the field. However, I have a few comments that might be addressed in a revision:

Response: We thank the reviewer for the positive evaluation and the helpful comments.

Comment 2: To me the most important aspect addressed in the paper is the role of language and communication on this question. I would also like to see specific measures discussed, like LIWC (Pennebaker).

Response: We thank the reviewer for this suggestion. We added the following sentences on page 8:

“The idea that the use of swear words and insults can be an indicator of hostility is based on the assumption that written text like social media comments reflect the cognitive and affective processes of the author. This assumption is widely shared across psychological and communication research and provides the basis for research on psycholinguistic dictionaries⁴⁶. For example, textual analysis programs like Linguistic Inquiry and Word Count (LIWC) have been used to categorise whether public discussions are dominated by hostile or civil language^{46,47}.”

Comment 3: Also hostility is manipulated by inappropriate words. To me, this is something different that hostile. Please explain in more depth why you opt for this manipulation (and report this in the limitations).

Response: Following this comment we added the following new paragraph to the Limitations section on page 32:

“Hostility can be defined in terms of its cognitive (e.g., cynical beliefs and distrust towards others), affective (e.g., anger and disgust) and behavioural components (e.g., aggressive physical behaviour and verbal aggression)^{70,71}. We chose to focus on the verbal expression of hostility in this study because it is common in social media discussions about polarized scientific issues⁹⁻¹¹. This focus also means that findings cannot be generalised to other expressions of hostility. For example, if actual physical violence or more subtle non-verbal face expressions of anger and disgust are part of a persuasive episode then this may impact the persuasiveness of individuals to a different extent than the use of verbal cues.”

Comment 4: The two foci “science denier” and “advocate for science” need explanation. Is an advocate for science a scientist. On the word level an symmetry between these two positions seem to be indicated. However, I would like to see a more thorough reflection for both positions. In doing so, I

would like to see a short passage on insights on deniers as such (see Laundry, <http://www.asheleylandrum.com>).

Response: We thank the reviewer for this suggestion and the reference. Following this comment we have added the following paragraph in the Introduction section on page 2:

“Science deniers reject scientific evidence because it poses a threat to their economic, social, or psychological interests^{3,4}. For example, spreading doubts about climate change can ensure fossil fuel sales, sharing the beliefs of a flat earth society can satisfy a need for belongingness, and refusing all vaccines or GMOs can be a strategy to cope with fears of needles or new technologies. In the face of scientific evidence, science deniers employ a variety of different strategies to justify their denial. They cite fake experts, conduct logical fallacies, raise impossible expectations, cherry pick data and construe conspiracy theories^{5,6}. These strategies are also often used in misinformation to persuade others⁷. Exposure to misinformation that denies science can reduce the publics’ positive attitudes toward behaviours dismissed by science deniers (e.g., vaccination) and the intention to perform these behaviours^{7,8}. Advocates for science can mitigate the damage of science denialism by rebutting misinformation in public discussions, for example, on social media⁹⁻¹². Advocates for science “follow scientific consensus and argue for the evidence-based position”⁹. They engage in science communication and rebuttal of misinformation either as professionals (e.g., scientists, science journalists) or as knowledgeable laypeople (e.g., amateur fact-checkers).”

Comment 5: Counterarguing and Advocacy are two concepts that fit the theoretical underpinning of the studies. Please elaborate more on these.

Response: We provided some more information on this on page 5:

In these discussions, the denier shares misinformation while the advocate counterargues against the misinformation by uncovering what is misleading about the misinformation (i.e., refutational messaging) and by providing additional scientific facts⁹. By engaging in this form of rebuttal the advocate advocates for the scientific perspective (e.g., the safety of vaccination).

Comment 6: From vaccination to gen-manipulated food is a big step. I see that the aim is to generalise the findings of Exp 3. However, a more thorough discussion on the role of topic is needed.

Response: Following this comment we added a more detailed explanation of the choice of topics in this study on page 5:

“Vaccination and GMOs are highly promising technologies to fight major threats to global public health such as infectious diseases and malnutrition. While scientists argue that risks and benefit analyses are crucial for each vaccine and GM food product, science deniers reject both technologies a priori and fuel general vaccine hesitancy and fears about GMOs in the public^{1,6,9}. Identifying effective interventions against misinformation that work across topics is useful to support the work of advocates for science. In addition, testing interventions across these topics provides an actual challenge for the generalisability of hypothesized effects because both topics have several unique aspects for the individual decision maker.”

Comment 7: Minors: Try to use a balanced wording (e.g. skip “fortunately” on page 2, line 23)

Response: We deleted the term fortunately and checked the manuscript for unbalanced wording.

27th Oct 23

Dear Philipp,

Your manuscript titled "The role of hostility when rebutting science denialism in social media discussions" has now been seen by our reviewers, whose comments appear below. In light of their advice I am delighted to say that we are happy, in principle, to publish a suitably revised version in Communications Psychology under the open access CC BY license (Creative Commons Attribution v4.0 International License).

We therefore invite you to revise your paper one last time to address the remaining concerns of our reviewers and a list of editorial requests. At the same time we ask that you edit your manuscript to comply with our format requirements and to maximise the accessibility and therefore the impact of your work.

Please note that it may still be possible for your paper to be published before the end of 2023, but in order to do this we will need you to address these points as quickly as possible so that we can move forward with your paper.

EDITORIAL REQUESTS:

I included a number of requests in particular related to the display items and SI. I'd encourage you to get in touch with any questions or concerns about any of our requests, so that these issues don't lead to unnecessary delays later on.

SUBMISSION INFORMATION:

OPEN ACCESS:

Communications Psychology is a fully open access journal. Articles are made freely accessible on publication under a [CC BY license](http://creativecommons.org/licenses/by/4.0) (Creative Commons Attribution 4.0 International License). This license allows maximum dissemination and re-use of open access materials and is preferred by many research funding bodies.

For further information about article processing charges, open access funding, and advice and support from Nature Research, please visit <https://www.nature.com/commspsychol/article-processing-charges>

At acceptance, you will be provided with instructions for completing this CC BY license on behalf of all authors. This grants us the necessary permissions to publish your paper. Additionally, you will be asked to declare that all required third party permissions have been obtained, and to provide billing information in order to pay the article-processing charge (APC).

* TRANSPARENT PEER REVIEW: Communications Psychology uses a transparent peer review system. On author request, confidential information and data can be removed from the published reviewer reports and rebuttal letters prior to publication. If you are concerned about the release of confidential data, please let us know specifically what information you would like to have removed. Please note that we cannot incorporate redactions for any other reasons.

* CODE AVAILABILITY: All Communications Psychology manuscripts must include a section titled "Code Availability" at the end of the methods section. We require that the custom analysis code supporting your conclusions is made available in a publicly accessible repository at this stage; please choose a repository that generates a digital object identifier (DOI) for the code; the link to the repository and the DOI must be included in the Code Availability statement. Publication as Supplementary Information will not suffice.

* DATA AVAILABILITY:

[link redacted]

Best regards,

Marike

Marike Schiffer, PhD
Chief Editor
Communications Psychology

REVIEWERS' COMMENTS:

Reviewer #1 (Remarks to the Author):

The authors have adequately addressed all the points raised during the initial review. I am satisfied with the revisions made to the manuscript, and I believe it is now ready for publication.

Reviewer #2 (Remarks to the Author):

Thank you for the in-depth revisions. My initial review of the paper was already very positive and I see all points addressed appropriately. I suggest acceptance.